# A snapshot of HIV-1 genetic diversity in Dominican Republic in 2024: Predominance of the B$_{Car}$ lineage and first description of a CRF02-AG isolate

Lily Soto[1], Yoneira F. Sulbaran[2], Rossana C. Jaspe[2], Francisco Ureña[3], Yoel Garcia[4], Carmen Luisa Loureiro[2], Héctor R. Rangel[2]*, Flor H. Pujol [2]*

1 Instituto de Medicina Tropical, UCV, Caracas, Venezuela, 2 Laboratorio de Virología Molecular, CMBC, IVIC, Caracas, Venezuela, 3 Hospital Regional Universitario Dr. Luis Manuel Morillo King, Especialista en Infectología, La Vega, República Dominicana, 4 Hospital Regional Presidente Estrella Ureña, Santiago, República Dominicana

☉ These authors contributed equally to this work.
* hrangel2006@gmail.com (HRR); fhpujol@gmail.com (FHP)

## Abstract

Although a decrease in the incidence of HIV-1 infection has been observed in the Caribbean, the prevalence of this viral infection is still around 1% in Dominican Republic. It has been shown that the subtype B circulating in many of the Caribbean islands is indeed a subtype B which was introduced before the expansion of the B Pandemic lineage (B$_{Pandemic}$), which has been called the Caribbean B lineage (B$_{Car}$). The aim of this descriptive study was to provide a snapshot of the present circulation of HIV-1 subtypes and lineages in this island. Different regions of the HIV-1 genome were amplified by RT-PCR and sequenced for phylogenetic analysis of the viral isolates circulating in La Vega and Santiago de los Caballeros, Dominican Republic, in 2024. Of the 26 HIV-1 isolates, all but one belonged to the HIV-1 subtype B. Several genomic regions were evaluated for their suitability for the lineage assignment. In addition to the PR/RT region, which was previously shown to discriminate between the two lineages, the INT region also allowed the discrimination, but VIF with less performance. In contrast, the NEF region did not seem suitable for lineage discrimination. From the 25 HIV-1 subtype B isolates, the lineage could be assigned in 20, 17 of them (85%) belonging to the B$_{Car}$ lineage. The other HIV-1 isolate was found to be a CRF02-AG isolate. The complete genome showed that this isolate was related to isolates circulating in Africa and Europe.

## Introduction

Although a sharp decrease in the incidence of HIV-1 infection has been observed in the Caribbean, the prevalence of this viral infection is still around 1% in Dominican

**Data availability statement:** All relevant data are within the paper and its Supporting information files.

**Funding:** MinCyt.

**Competing interests:** The authors have declared that no competing interests exist.

Republic (DR), and 4% in its neighboring country Haiti [1]. Around 84,000 people living with HIV-1 (PLWH) were estimated in DR in 2023 [2].

HIV-1 is a member of the *Retroviridae* family, genus *lentivirus*, which is divided into four groups (M, N, O, and P), each of which being the product of an independent spillover event from simian viruses infecting non-human primates (chimpanzees and/or gorillas and recombinant viruses from simian viruses infecting those primates) [3]. The Main group M is the one responsible for the Pandemic and has evolved to diversify into 10 subtypes (A-D,F-L) and almost 200 circulating recombinant forms (CRF), strain for which at least three isolates have been identified: product of the recombination of at least two subtypes, in addition to unique recombinant forms, one isolate product of recombination [4–6].

HIV-1 subtype B is predominant in all the Americas [7,8] and in most Caribbean islands [9,10], although other subtypes and recombinant forms are more frequent in Brazil and the South Cone [11,12]. It has been shown that the subtype B circulating in many Caribbean islands is indeed a subtype B introduced before the expansion of the B Pandemic lineage ($B_{Pandemic}$): this lineage has been called the Caribbean B lineage ($B_{Car}$) [9]. In DR, the $B_{Car}$ lineage has been predominant [9,13]. However, the HIV-1 genetic diversity is a dynamic process in each country, particularly influenced by migration fluxes and tourism. The main origin of tourists in the DR is the United States and Canada, forming North America's dominant source market, with significant numbers also coming from South American countries like Argentina and Colombia, and from Europe (Spain and France) [14]. In all these countries, the $B_{Car}$ lineage is not predominant, and in the two countries (USA and Canada) that contribute with the most abundant tourists in DR, although HIV-1 $B_{Pandemic}$ is predominant, several other subtypes and recombinant forms have been identified [15].

The last HIV-1 sequences of DR available at LANL are from 2018. This study aims to provide a snapshot of the present circulation of HIV-1 subtypes and lineages in DR.

## Materials and methods

### Population group

A total of 26 plasma samples from patients living with HIV-1 in La Vega and Santiago de los Caballeros, DR, and collected in June 2024, were analyzed in this study. All but four patients were Naïve upon blood collection and were then proposed for antiretroviral treatment. Viral load (when available) ranged between 50–1100000. A written informed consent was obtained from all the patients. This study was approved by the Bioethical Committee of Consejo de Enseñanza, La Vega, Dominican Republic (04/2024: CBE N 01/2024), and Hospital Universitario de Caracas, UCV, Venezuela (CBE 08/2024).

### Amplification and sequencing

Sequencing of the HIV-1 complete genome or another important region like the PR/RT region, is not always feasible, depending on the viral load of the patient and the

integrity of the samples upon arrival to the laboratory. Since the viral load was not high for all the plasma samples and some of them suffered with the transportation to the laboratory, other regions, usually easier to amplify, were evaluated, to assess their suitability for determining the HIV-1 subtype and the lineage ($B_{Car}$ or $B_{Pandemic}$).

The following regions were assessed for amplification and sequencing, according to previously reported procedures: the PR/RT, INT, NEF, and VIF regions [8,16]. In addition, the complete HIV-1 genome was produced for one sample by Next Generation Sequencing, using the procedure described above. Viral RNA was extracted using QIAamp Viral RNA Mini kit (Qiagen, Hilden, Germany). Reverse transcription of RNA and PCR were performed using SuperScript™ IV One-Step RT-PCR System (Invitrogen, USA), to produce two overlapped amplicons, one of 5.2 kb and the other of 4.6 kb [17]. The first half of the genome, genes gag to vpu, was amplified with Gag3_For (positions 785–803) and Vpu3_Rev (positions 5978–5956) primers. The other half, Vif to Ltr genes, was amplified with Vif3_For (positions 5037–5060) and OFM19 (positions 9632–9604) primers. The PCR mix consisted of 2.8 µL nuclease-free water, 7.5 µL 2X buffer, 0.2 µL Super-Script™ IV One-Step RT-PCR System, 0.75 µL forward and reverse primers, and 3 µL ARN, in a total reaction volume of 15 µL, with the following PCR conditions: an incubation at 55 °C for 30 min, followed by 98 °C/2 min and 40 cycles of 94 °C/10 s, 55 or 63 °C/20 s and 72 °C/30 s per kb, with a final extension of 72 °C for 7 min. These amplicons cover the entire genome; equal amounts of each amplicon were used to generate a pool subjected to next-generation sequencing (NGS) using the Illumina platform and the Microbial Amplicon Prep Kit (iMAP) (Illumina, Inc., San Diego, CA, USA), using IDT for Illumina-PCR Indexes set 1 following the manufacturer's recommendations. The libraries were pooled and quantified (Qubit DNA HS, Thermo Scientific, Waltham, MA, USA). The sequencing was performed with 10% PhiX control v3, using an iSeq 100 platform, a loading concentration of 50pM, and a 300-cycle V2 kit with paired-end sequencing. Viral genome assembly was performed using the Genome Detective tool (https://www.genomedetective.com/db/ui/login). This application also assigned a subtype for each isolate. The nucleotide sequence data were deposited into the GenBank database under the accession numbers PX989794-PX989838.

## Phylogenetic analysis

Sequences were aligned with reference sequences for $B_{Car}$ and $B_{Pand}$ [13], allowing inclusion of sequences from isolates from different countries. For the CRF02-AG isolate, in addition, a Blast analysis was performed for both the complete genome sequence and the PR/RT region to include the most similar sequences in each phylogenetic tree. The alignments were performed using MAFFT [18] and edited manually using MEGA version 12 [19]. The phylogenetic trees were performed by the Maximum Likelihood method, using IQ-Tree [20] and edited using ITol version 7.4.2 [21]. Bootstrap values over 70 were considered to define a highly supported clade.

The analysis of the recombination pattern was performed for the sequence HIV1113 by using the recombination identification program (RIP) [15]. HIV1113 and other selected samples were loaded into the RIP and analyzed using A1 and G consensus sequences as references. The genetic map with the recombination points was obtained for each sequence analyzed.

## Results

### Subtype and lineage analysis

A total of 45 sequences were analyzed in this study: four from the PR/RT region, five from the INT gene, 18 for the VIF and 17 for the NEF genes, and one complete genome, from 26 HIV-1 isolates from Dominican Republic (Table 1). All but one of these isolates belonged to HIV-1 subtype B, while the complete genome sequence corresponded to a CRF02-AG recombinant form.

Different genomic regions were evaluated for their suitability for differentiating the two lineages, $B_{Car}$ and $B_{Pandemic}$. With the complete genome and the PR/RT sequences, the two subtype B lineages formed two distinct clades, as described

**Table 1. Sequences available from the 28 HIV-1 patients from this study.**

| Isolate[1] | PR/RT | INT | VIF | NEF |
|---|---|---|---|---|
| HIV1079 | | | BCar | B |
| HIV1081 | BCar[3] | BCar | BCar | BCar? |
| HIV1082 | BCar | BCar | BCar | BCar? |
| HIV1083 | | | BPand | |
| HIV1084 | | | BCar | B |
| HIV1086 | | | BCar | |
| HIV1087 | | | | B |
| HIV1088 | | | BCar | B |
| HIV1089 | BCar | BCar | BCar | B |
| HIV1090 | | BCar | | B |
| HIV1094 | | | | B |
| HIV1095 | | | BCar | BCar? |
| HIV1096 | | | Bpand | |
| HIV1102 | | | BCar | |
| HIV1103 | | | BCar | |
| HIV1105 | | | | B |
| HIV1106 | | | | B |
| HIV1108 | | | BCar | BCar? |
| HIV1109 | | | BCar | |
| HIV1111 | | | | BCar? |
| HIV1112 | | | BCar | |
| HIV1114 | BCar | BCar | | |
| HIV1115 | | | BCar | BCar? |
| HIV1116 | | | BCar | B |
| HIV1117 | | | BCar | B |
| HIV1113 | CRF02-AG[4] | | | |

[1]All the isolates were from Naïve patients, except for patients 1090, 1102, 1108 and 1114 who were on treatment withdrawal. [2]NA: not available. [3]BCar: $B_{Car}$. BPand: $B_{Pandemic}$, assigned according to the phylogenetic analysis of Fig 1. For the NEF sequence, the lineage could not be inferred for many isolates. [4]Complete genome sequence was available for this isolate.

previously [13] (S1 Fig in S1 File and Fig 1A). The four sequences from the PR/RT region were grouped into the $B_{Car}$ lineage (Fig 1A). The phylogenetic analysis of the INT region did not allow a sharp, clear discrimination between the two lineages, compared to the PR/RT region. However, the two clades were clearly differentiated. The five sequences from the INT region were grouped into the $B_{Car}$ lineage (Fig 1B). The phylogenetic analysis of the VIF region did not allow a differentiation between the two lineages as clear as for the previous regions mentioned above. However, most of the sequences from the $B_{Car}$ lineage were grouped into a clade. Other sequences from the same lineage formed independent clades inside the $B_{Pandemic}$ lineage clade (Fig 1C). According to their grouping, all but two (these two isolates HIV1083 and HIV1096, closely related) of the sequences analyzed in this study seemed to belong to the $B_{Car}$ lineage (Fig 1C). The phylogenetic analysis of the NEF region did not allow a clear discrimination between the two lineages: for many of the sequences the lineage could not be assigned (Fig 1D) (Table 1). Therefore, the assignment provided by the phylogenetic analysis of this region was not taken into con consideration for the lineage assignment. The analysis of the PR/RT, INT, and, with limitations, VIF sequences, allowed to assign the lineage of 20 out of the 25 subtype B isolates: 17 of them were classified as $B_{Car}$ lineage (85%).

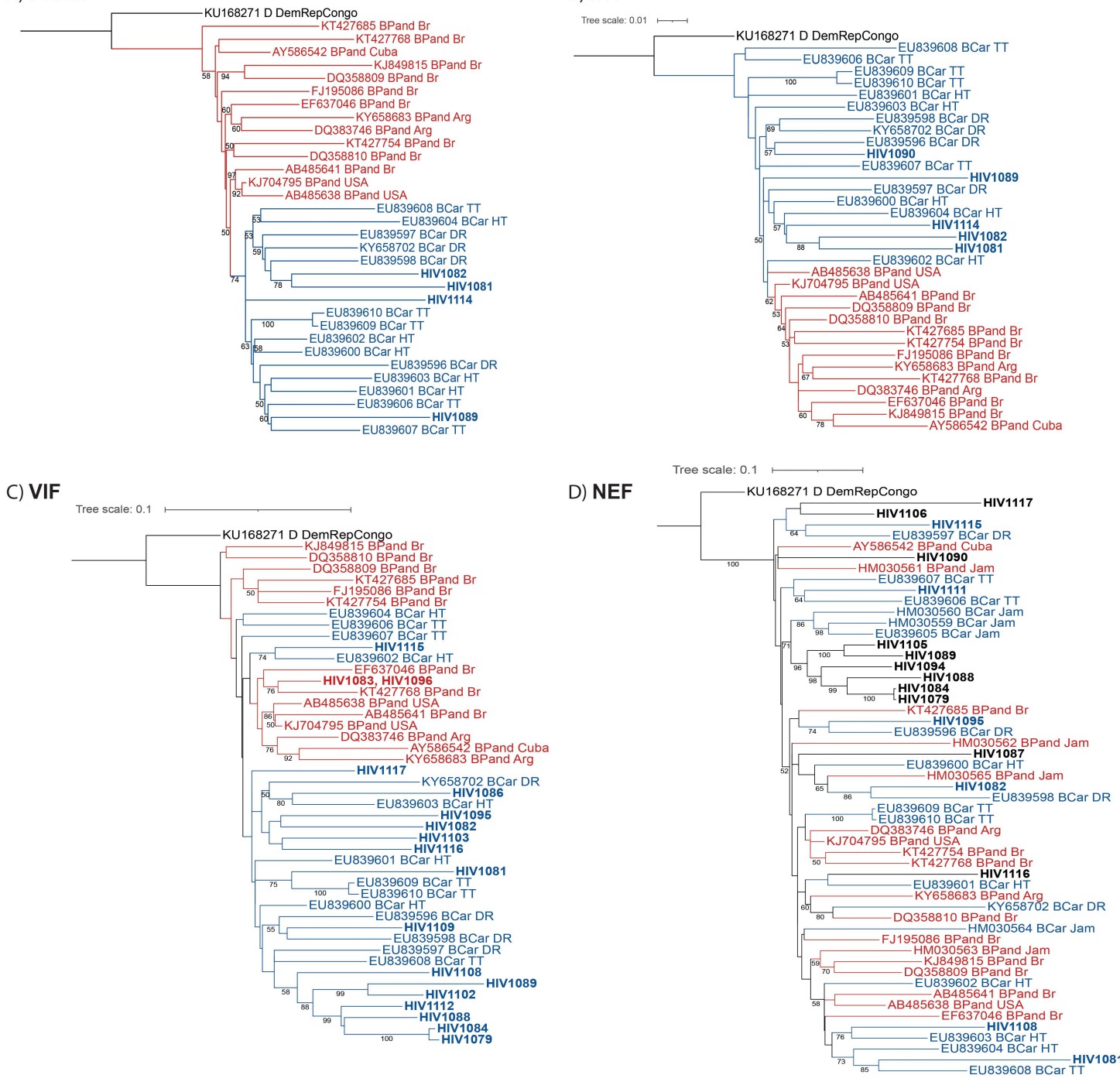

**Fig 1. Phylogenetic analysis of different genomic regions of HIV-1 for discrimination of the B$_{Car}$ and B$_{Pandemic}$ lineage.** Phylogenetic inference was performed by the Maximum Likelihood method, with 1000 bootstrap value, using the substitution model described in S1 Table in S1 File. 1A to 1D: phylogenetic tree of PR/RT, INT, VIF and NEF region respectively. The B$_{Car}$ isolates are shown in blue and the B$_{Pandemic}$ ones in red. The accession numbers are shown for each isolate, then the lineage (or subtype different than B) for the reference sequences and the country (Argentina, Brazil, Cuba, Democratic Republic of Congo, DR, Haiti, Trinidad and Tobago, USA). More B$_{Car}$ sequences (from Jamaica) were used for the phylogenetic tree of the NEF region since many of the B$_{Car}$ region did not cover the complete coding NEF region. The accession numbers of the reference sequences are included in S2 Table in S1 File.

## Analysis of the CRF01-AG isolate

The complete genome of a CRF02-AG was also identified among the 26 HIV-1 isolates (Fig 2). The sequence branched with three sequences: AY093604 (Senegal), AY093605 (West Africa), and AY093607 (Germany), although it was not closely related. When only the PR/RT region was analyzed, the sequence was grouped with the two sequences displaying higher identity on Blast analysis: AJ286136 (Senegal) and EU248477 (Belgium), which belong to the lineage of the Ibng prototype CRF02-AG (Accession L39106) (S2 Fig and S4 Table in S1 File).

The recombination pattern was analyzed for the HIV1113 sequence and for two of the sequences which were grouped with it (the third sequence was shorter and was not included). The recombination pattern of the DR isolate was somewhat similar to those sequences compared to the recombination pattern of the reference sequences (Fig 3). However, it was not completely similar. This may be associated with the fact that the HIV1113 was not closely grouped with those sequences (Fig 2).

## Discussion

The first HIV-1 case was reported in 1983 in DR [22]. Several pieces of evidence suggest that Haiti was the first country of the Americas where HIV-1 was introduced around 1960, leading to the $B_{Car}$ lineage, anterior to the $B_{Pandemic}$ one [9]. The

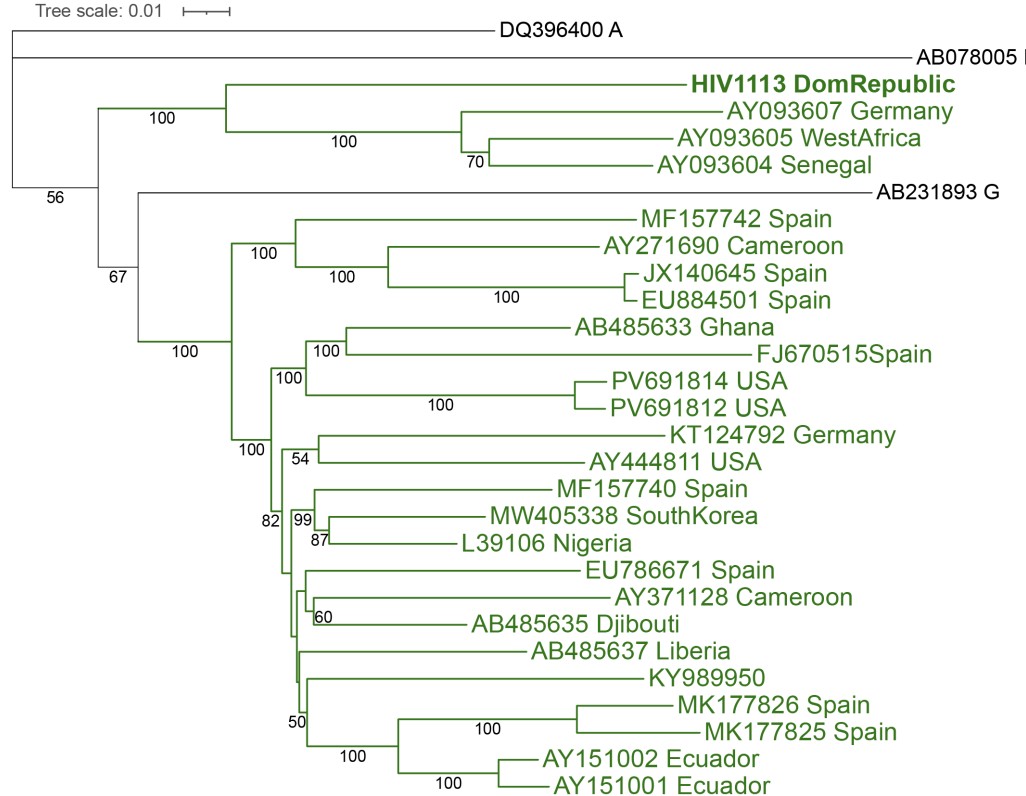

**Fig 2. Phylogenetic analysis of the near complete genome of the CRF02-AG isolate from DR.** Phylogenetic inference was performed by the Maximum Likelihood method, with 1000 bootstrap value, using the substitution model described in S1 Table. The accession number and the country are shown for each isolate. The CRF02-AG sequences are shown in green. For the other reference sequences, the subtype is shown after the accession number (see S3 Table n S1 File).

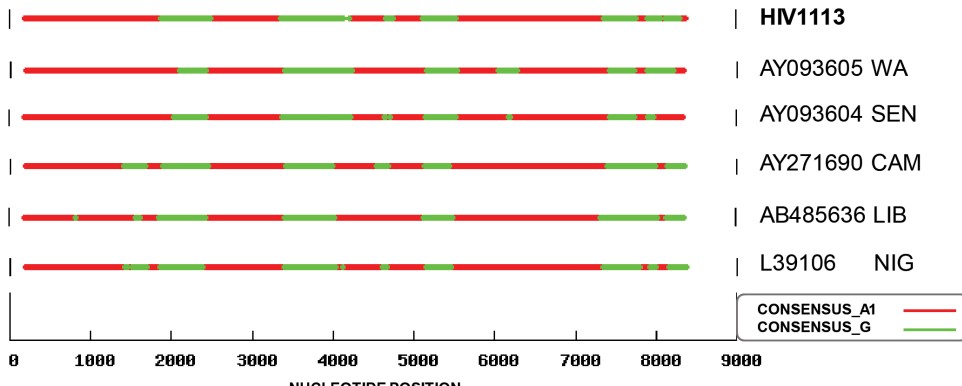

**Fig 3. Genetic map of the recombination points of the CRF02-AG isolate from DR.** The recombination pattern of the DR isolate was compared with the sequences that were grouped in the same clade (AY093605 from West Africa and AY093604 from Senegal) and the three reference sequences for the CRF02-AG (AY271690 from Cameroon, AB485636 from Liberia and L39106 from Nigeria).

$B_{Car}$ lineage has been shown to predominate in DR [9,13]. However, since, several years have passed since the last HIV-1 sequences were analyzed, the purpose of this study was to evaluate whether the $B_{Car}$ lineage was still predominant in the island.

In their previous studies on the BCar lineage, the complete genome, the PR/RT region or the ENV one was used for the lineage assignment [13]. Other regions were analyzed in this study to assess their suitability for the lineage assignment. Although they provided a less optimal discrimination than, for example, the PR/RT region, the phylogenetic analysis of the INT gene and even the VIF gene allowed to suggest that the $B_{Car}$ lineage was still predominant in DR. In contrast, the NEF region did not prove to be suitable for this lineage discrimination.

Even with the low number of isolates analyzed in this study, the analysis also allowed to identify a CRF: CRF02-AG. This recombinant form is one of the most common CRFs in the world [23,24] and is predominant in West and Central Africa [23]. Recombinant forms have already been reported in DR: one isolate BC out of 18 samples from DR (collection date 2005) [25], one isolate BF (collection date 2018) among HIV-1 isolates infecting 50 female sex workers from DR [26]. These two sequences are available at GenBank out of around 2400 sequences (mostly subtype B) available for the country until 2024. For the BC recombinant, almost the complete genome sequence was deposited in GenBank. Another study identified three HIV-1 isolates with a recombination pattern like the BC recombinant reported in DR, suggesting that this CRF might be circulating in the Caribbean [27].

No CRF02-AG was reported previously in DR. In the Americas, the countries with the most sequences of this CRF are the USA and Canada, although most isolates are from Africa. However, this CRF has been circulating in both countries [28,29]. The CRF02-AG has also been identified in Brazil [30], Mexico [31], Panama [32], and Ecuador [33]. This CRF is the most common non-B subtype in Guadeloupe and particularly Martinique and has been described in French Guiana [34]. The CRF02-AG found in DR did not appear to be related to the CRF02-AG circulating in the Americas and the Caribbean: it clustered with sequences from Africa and Europe.

A limitation of this study is the low number of HIV-1 analyzed. However, even with this limitation, the analysis allowed to suggest that the $B_{Car}$ is still predominant in DR, although recombinant forms appear sporadically in the island. Our data, although limited, warn of a possible dynamic of introduction of new subtypes and/or CRF to DR, suggesting the necessity of more active surveillance studies not only in DR but in all the Americas. More studies are needed to evaluate the circulation of other subtypes and particularly the frequency of this CRF02-AG lineage in the island.

## Conclusions

In conclusions, the phylogenetic analysis of several HIV-1 genomic regions allows to suggest that the $B_{Car}$ lineage is still predominant in DR, although the recombinant forms appear sporadically in the country. Although this study provides a snapshot of current HIV-1 diversity in two urban centers in DR, it is based on a relatively small sample. Therefore, the observed predominance of the $B_{Car}$ lineage and the detection of a single CRF02-AG isolate should not be extrapolated to the entire country without caution. Larger, systematically sampled studies that include additional geographic regions and risk groups are needed to accurately appraise the HIV-1 genetic diversity in DR. Nevertheless, our findings highlight the ongoing introduction of new lineages into DR and underscore the importance of continued molecular surveillance in the Caribbean region.

## Supporting information

**S1 File. S1 Table.** Information data of the phylogenetic trees. **S2 Table**. Sequences used for subtype B lineage. **S3 Table**. Complete genome sequences used for subtype CRF02-AG. **S4 Table.** Accession numbers of the sequences of PR/RT region used for subtype CRF02-AG. **S1 Fig.** Phylogenetic analysis of the complete genome of HIV-1 for discrimination of the BCar and BPandemic lineage. **S2 Fig.** Phylogenetic analysis of the PR/RT region of HIV-1 CRF02-AG. (ZIP)

## Acknowledgments

This work was supported by funding from MinCyT, Venezuela.

## Author contributions

**Conceptualization:** Lily Soto, Yoneira F Sulbaran, Rossana C. Jaspe, Hector R. Rangel, Flor H. Pujol.

**Data curation:** Carmen Luisa Loureiro, Flor H. Pujol.

**Formal analysis:** Rossana C. Jaspe, Carmen Luisa Loureiro.

**Investigation:** Lily Soto, Yoneira F. Sulbaran, Rossana C. Jaspe, Hector R. Rangel, Flor H. Pujol.

**Methodology:** Lily Soto, Yoneira F. Sulbaran.

**Resources:** Francisco Urena, Yoel Garcia.

**Supervision:** Flor H. Pujol.

**Validation:** Hector R. Rangel.

**Writing – original draft:** Flor H. Pujol.

**Writing – review & editing:** Lily Soto, Yoneira F. Sulbaran, Rossana C. Jaspe, Francisco Urena, Yoel Garcia, Carmen Luisa Loureiro, Hector R. Rangel, Flor H. Pujol.

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
