## [Decision Letter · Decision Letter 0]

24 Mar 2026

PONE-D-26-05473A snapshot of HIV-1 genetic diversity in Dominican Republic in 2024: predominance of the BCar lineage and first description of a CRF02-AG isolatePLOS One

Dear Dr. Pujol,

Thank you for submitting your manuscript to PLOS ONE. After careful consideration, we feel that it has merit but does not fully meet PLOS ONE’s publication criteria as it currently stands. Therefore, we invite you to submit a revised version of the manuscript that addresses the points raised during the review process.

The authors have shown solid evidence and achieved compelling conclusions. Minor concerns arise from the need to broaden the descriptions of methods and results and to delve deeper into the data discussion.

We look forward to receiving your revised manuscript.

Kind regards,

Elisabetta Pilotti

Academic Editor

PLOS One

Journal Requirements:

“This work was supported by funding from MinCyT, Venezuela.”

“MinCyt”

“MinCyt”

4. Please note that your Data Availability Statement is currently missing the DOI/accession number of each dataset OR a direct link to access each database. If your manuscript is accepted for publication, you will be asked to provide these details on a very short timeline. We therefore suggest that you provide this information now, though we will not hold up the peer review process if you are unable.

5. Please amend the manuscript submission data (via Edit Submission) to include author Francisco Urena.

6. Please amend your authorship list in your manuscript file to include author Francisco Ureña.

7. We notice that your supplementary figures are uploaded with the file type 'Figure'. Please amend the file type to 'Supporting Information'. Please ensure that each Supporting Information file has a legend listed in the manuscript after the references list.

Reviewers' comments:

Reviewer's Responses to Questions

Comments to the Author

1. Is the manuscript technically sound, and do the data support the conclusions?

Reviewer #1: Partly

Reviewer #2: Yes

2. Has the statistical analysis been performed appropriately and rigorously? 

Reviewer #1: No

Reviewer #2: N/A

3. Have the authors made all data underlying the findings in their manuscript fully available?

Reviewer #1: No

Reviewer #2: Yes

4. Is the manuscript presented in an intelligible fashion and written in standard English?

Reviewer #1: Yes

Reviewer #2: Yes

5. Review Comments to the Author

Reviewer #1: In this study, Soto et al provide a snapshot of the present (2024) circulation of HIV-1 subtypes and lineages in the Dominican Republic. The authors observed that the subtype B, and particularly the BCAR lineage, is still predominant in Dominican Republic, although recombinant forms appear sporadically in the island. This study provides novel and interesting information about current HIV-1 genetic diversity in the Caribbean region. Some data, however, should be better described and discussed in order to provide more solid conclusions.

1) The authors must describe the criteria used to select the reference sequences for determining the subtype and lineage of each fragment via phylogenetic analyses. Additionally, the accession numbers of all subtype B and CRF02_AG reference sequences should be provided as supplementary data.

2) The authors should describe the statistical test of phylogenetic branch support and the threshold use to define a highly supported clade.

3) While the authors distinguish between the POL and INT regions, it is more appropriate to refer to the first region as PR/RT rather than POL, as the integrase (int) gene is actually part of the polymerase (pol) coding region.

4) The authors state that for the VIF region, most of the sequences from the BCAR lineage were grouped into a clade, while other sequences from the same lineage formed independent clades inside the BPANDEMIC lineage. However, even the BCAR clade identified did not achieve significant branch support. Consequently, neither this region nor NEF enabled a clear discrimination between the two subtype B lineages and should not be used to discriminate between them.

Reviewer #2: The authors need to make minor changes for possible publication. Authors must make minor revisions for potential publication. Authors must specify the ethics approval number and the name of the ethics committee. The remaining comments can be found in the file titled "Reviewer comments".

6. PLOS authors have the option to publish the peer review history of their article (what does this mean?). If published, this will include your full peer review and any attached files.

Do you want your identity to be public for this peer review? For information about this choice, including consent withdrawal, please see our Privacy Policy.

Reviewer #1:  Yes: Gonzalo Bello

Reviewer #2: No

---

## [Author Response · Author response to Decision Letter 1]

10 Apr 2026

Caracas, April 4, 2026

Dr. Yuri Khudyakov

Editor. PLos ONE

Dear Dr. Khudyakov:

Please find enclosed our revised version of our manuscript entitled: “A snapshot of HIV-1 genetic diversity in Dominican Republic in 2024: predominance of the BCar lineage and first description of a CRF02-AG isolate" by Lily Soto, Yoneira F Sulbaran, Rossana C Jaspe, Francisco Ureña, Yoel Garcia, Carmen Luisa Loureiro, Héctor R Rangel, and Flor H. Pujol, to be considered for publication in PLOS ONE. The paper consists of 17 pages, one table, 3 figures, 4 supplementary tables and 2 supplementary figures. This manuscript has not been published previously and is not under consideration for publication elsewhere.

We have revised the manuscript according to the constructive comments of the reviewers. A revised version, with the edition highlighted in red, is also provided.

Looking forward to hearing from you soon. Sincerely,

Flor Helene Pujol, Professor, Head.

Lab. de Virología Molecular

CMBC. IVIC. Apdo 20632

Caracas 1020-A. Venezuela

e-mail: fhpujol@gmail.com

Reviewer #1: Dr. Gonzalo BELLO

In this study, Soto et al provide a snapshot of the present (2024) circulation of HIV-1 subtypes and lineages in the Dominican Republic. The authors observed that the subtype B, and particularly the BCAR lineage, is still predominant in Dominican Republic, although recombinant forms appear sporadically in the island. This study provides novel and interesting information about current HIV-1 genetic diversity in the Caribbean region. Some data, however, should be better described and discussed in order to provide more solid conclusions.

Thank you for your comment. We hope to have addressed these comments adequately.

1) The authors must describe the criteria used to select the reference sequences for determining the subtype and lineage of each fragment via phylogenetic analyses. Additionally, the accession numbers of all subtype B and CRF02_AG reference sequences should be provided as supplementary data.

The reference sequences were selected allowing the inclusion of sequences from isolates from different countries. For the CRF02-AG isolate, in addition, a Blast analysis was performed for both the complete genome sequence and the POL region to include the most similar sequences in each phylogenetic tree. This information was included in lines 120-125. The list of Accession numbers is now included in S2-4 Tables as suggested.

2) The authors should describe the statistical test of phylogenetic branch support and the threshold use to define a highly supported clade.

Bootstrap values (statistical test of phylogenetic branch support) over 70 were considered to define a highly supported clade. This information was included in line 128.

3) While the authors distinguish between the POL and INT regions, it is more appropriate to refer to the first region as PR/RT rather than POL, as the integrase (int) gene is actually part of the polymerase (pol) coding region.

Thank you for your comment. POL was substituted by PR/RT throughout the text as suggested.

4) The authors state that for the VIF region, most of the sequences from the BCAR lineage were grouped into a clade, while other sequences from the same lineage formed independent clades inside the BPANDEMIC lineage. However, even the BCAR clade identified did not achieve significant branch support. Consequently, neither this region nor NEF enabled a clear discrimination between the two subtype B lineages and should not be used to discriminate between them.

Thank you for your comment. The limitations of the VIF region for assigning the lineage were included in lines 157-159, line 163 and line 168.

REVIEWER #2: Dr. Christian MANGALA

The authors need to make minor changes for possible publication. Authors must make minor revisions for potential publication. Authors must specify the ethics approval number and the name of the ethics committee. The remaining comments can be found in the file titled "Reviewer comments".

Thank you for your comments. The Ethics approval number was included for the DR document in line 83; it was already included for the Venezuelan document.

This manuscript is titled: ". A snapshot of HIV-1 genetic diversity in Dominican Republic

in 2024: predominance of the BCar lineage and first description of a CRF02-AG isolate",

address an important topic from the perspective of the molecular epidemiology of HIV-1 in the Dominican Republic in 2024. However, the manuscript presented by the authors requires minor revisions for improvement.

Thank you for your comment. We hope to have addressed the comments adequately.

Abstract

Please specify the type, location, and period of the study. Also specify the techniques used.

The type, location and period of the study, in addition to the techniques used, were included in Abstract, lines 29-32.

Introduction

Line 47: The authors must clearly specify the four groups, that is to say M, O, N and P.

The groups were specified in line 48, and the spill over events in lines 50-51.

Materials and Methods

- Authors must specify the type and location of studies.

The locations were specified in lines 77-78.

- Please provide the ethics clearance number issued by the ethics committee of the Dominican Republic.

The Ethics approval number was included for the DR document in line 83; it was already included for the Venezuelan document.

- The “Materials and Methods” section should contain subsections for better clarity.

The subsections were included.

- Please place the “Ethical Considerations” section after the “Conclusion” section.

The section was included (Lines 269-272).

Results

The “Results” section should include subsections for clarity.

Subsections were included in Results.

The figures are not of good quality; authors must submit legible and clear figures.

We apologize for supplemental Figure 2, for which the accession numbers are not legible because of the number of sequences. The list of these accession number is now included in Supplemental Table 4.

Conclusion

Authors should include a “Conclusion” section for clarity.

A Conclusion section was included as suggested (lines 256-267).

References

Authors must harmonize the list of references, taking into account the standard required by the journal (e.g., number of authors).

The references were edited in concordance to the guidelines.

---

## [Editor Report · Decision Letter 1]

14 Apr 2026

A snapshot of HIV-1 genetic diversity in Dominican Republic in 2024: predominance of the BCar lineage and first description of a CRF02-AG isolate

PONE-D-26-05473R1

Dear Dr. Pujol,

We’re pleased to inform you that your manuscript has been judged scientifically suitable for publication and will be formally accepted for publication once it meets all outstanding technical requirements.

Kind regards,

Elisabetta Pilotti

Academic Editor

PLOS One

Additional Editor Comments (optional):

I suggest the authors remove the adjective "descriptive" before "study", as written in the abstract at line 29.
---

## [Editor Report · Acceptance letter]

PONE-D-26-05473R1

PLOS One

Dear Dr. Pujol,

I'm pleased to inform you that your manuscript has been deemed suitable for publication in PLOS One. Congratulations! Your manuscript is now being handed over to our production team.

Kind regards,

on behalf of

Dr. Elisabetta Pilotti

Academic Editor

PLOS One